# On Determination of the Effective Thermal Conductivity of a Bundle of Steel Bars Using the Krischer Model and Considering Thermal Radiation

**DOI:** 10.3390/ma14164378

**Published:** 2021-08-05

**Authors:** Rafał Wyczółkowski, Vazgen Bagdasaryan, Stanisław Szwaja

**Affiliations:** 1Department of Production Management, Czestochowa University of Technology, Armii Krajowej 19, 42-200 Czestochowa, Poland; 2Institute of Civil Engineering, Warsaw University of Life Sciences—SGGW, Nowoursynowska, 166, 02-787 Warsaw, Poland; vazgen_bagdasaryan@sggw.edu.pl; 3Department of Thermal Machinery, Czestochowa University of Technology, Armii Krajowej 21, 42-200 Czestochowa, Poland; szwaja@imc.pcz.czest.pl

**Keywords:** heat treatment, bar bundle, effective thermal conductivity, Krischer model, thermal radiation, cellular medium

## Abstract

Cellular solid materials are commonly found in industrial applications. By definition, cellular solids are porous materials that are built of distinct cells. One of the groups of such materials contains metal foams. Another group of cellular metals contains bundles of steel bars, which create charges during the heat treatment of the bars. A granular structure connected by the lack of continuity of the solid phase is the main feature that distinguishes bundles from metal foams. The boundaries of the bundle cells are made of adjacent bars, with the internal region taking the form of an air cavity. In this paper, we discuss the possibility of using the Krischer model to determine the effective thermal conductivity of heat-treated bundles of steel bars based on the results of experimental tests and calculations. The model allows the *k_ef_* coefficient to be precisely determined, although it requires the weighting parameter *f* to be carefully matched. It is shown that the value of *f* depends on the bar diameter, while its changes within the examined temperature range (25–800 °C) can be described using a third-degree polynomial. Determining the coefficients of such a polynomial is possible only when the effective thermal conductivity of the considered charge is known. Moreover, we analyze a simplified solution, whereby a constant value of the *f* coefficient is used for a given bar diameter; however, the *k_ef_* values obtained thanks to this approach are encumbered with inaccuracy amounting to several dozen percentage points. The obtained results lead to the conclusion that the Krischer model cannot be used for the discussed case.

## 1. Introduction

Bundles of bars are examples of porous charges that can be encountered during the heat treatment of steel products [1]. Due to its cellular structure, a bundle of bars (an example of which is presented in Figure 1) is a granular two-phase medium, where one phase is steel and the other phase is gas, which fills in spaces that occur between individual bars. For this reason, the basic thermal property of a charge is the effective thermal conductivity *k_ef_*, which is commonly used in the theory of porous [2,3] and nonhomogeneous [4,5,6] media. Knowledge of the value of this coefficient is necessary to determine the heating time of the heat-treated bars [7,8].

The most reliable source of information on the effective thermal conductivity is experimental investigations. The effective thermal conductivity of cellular materials is most often measured via the steady-state method using a guarded hot plate apparatus [9,10,11]; however, this method of determining the *k_ef_* coefficient requires the use of specialist test equipment. Moreover, such measurements are time-consuming and require the preparation of suitable test samples; however, the main disadvantage of the measurements is that their results are not general, since they refer to strictly defined material samples. Therefore, while determining the effective thermal conductivity of cellular materials, in many cases it is common to use model calculations as effective alternatives for experimental investigations [12]. In the literature on the subject, various analytical models are used to determine the effective thermal conductivity of two-phase media (solid body-gas). These models can be divided into two basic categories (groups): rigid and flexible models. The rigid models present the value of the *k_ef_* coefficient only as a function of the thermal conductivities of the solid phase *k_s_* and the gas phase *k_g_*, as well as the medium porosity *ϕ*. The so-called flexible models, apart from the three enumerated values, take into account additional parameters, meaning they offer much wider possibilities. This group includes the Krischer model, among others. In this study, we investigated the possibility of applying this model to determine the thermal properties of heat-treated bundles of steel bars. The results of the experimental investigations were used to perform the necessary calculations.

## 2. Materials and Methods

The most common rigid models of effective thermal conductivity are the so-called structural models—the parallel model, the series model, the effective medium theory (EMT) model, and the Maxwell–Eucken model [13]. It has been shown that these models are not suitable for determining the effective thermal conductivity of bundles of bars [14]. The Krischer model (Equation (1)) mentioned in the Introduction, which is rated highly among elastic models, is the weighted harmonic mean of the parallel model (Equation (2)) and series model (Equation (3)) of the effective thermal conductivity [15]:(1)$$k_{ef}={\left[\frac{1-f}{k_{eP}}+\frac{f}{k_{eS}}\right]}^{-1},$$
where:(2)$$k_{eP}=\left(1-\varphi \right)k_{s}+\varphi k_{g},$$
(3)$$k_{eS}={\left(\frac{1-\varphi }{k_{s}}+\frac{\varphi }{k_{g}}\right)}^{-1}$$

When the weighting parameter *f* in Equation (1) assumes the value 0, a parallel model is obtained, which is the upper bound of the effective thermal conductivity; however, when *f* equals 1, a series model is obtained, which defines the lower bound of the effective thermal conductivity. By adjusting the *f* parameter properly, it is possible to obtain any value of the effective thermal conductivity that is in the range of the above-mentioned extremes; therefore, with the *f* parameter correctly fit, it is possible to obtain the correct value of the *k_ef_* coefficient for a given porous medium. Due to its great flexibility and relative simplicity, we decided to analyze the usefulness of the Krischer model in determining the thermal properties of heat-treated bundles of bars. In cases where it is easy to match the results, the Krischer model will offer ample application opportunities. A fundamental part of the performed calculations boils down to determining the value of the weighting parameter *f* from Equation (1), for which the *k_ef_* coefficient obtained with the use of the Krischer model will correspond to the real ability to transfer heat in the analyzed charge established by experimental investigations. The following section of the paper describes the method used to determine the *f* parameter for the cellular medium in the form of a round bar bundle.

The unknown *f* coefficient from the Krischer model was obtained from Equations (1)–(3) on the assumption that the value of the effective thermal conductivity of the analyzed medium *k_ef_* is known. After performing adequate transformations, eventually the following relationship was obtained (Equation (4)):(4)$$f=\left[\frac{\left(1-\varphi \right)k_{s}+\varphi k_{g}}{k_{ef}}-1\right]\cdot {\left[\frac{\left(1-\varphi \right)k_{s}+\varphi k_{g}}{{\left(\frac{1-\varphi }{k_{s}}+\frac{\varphi }{k_{g}}\right)}^{-1}}-1\right]}^{-1},$$

The effective thermal conductivity that occurs in Equation (4) was determined on the basis of the experimental measurements, which were performed using the stationary method in a guarded hot plate apparatus in single-side mode [16,17]. When testing consolidated media, the investigated samples are flat plates, whereas when testing porous media, such as bundles of round bars, the samples are flatbeds with a staggered arrangement. Figure 2 presents the scheme of the geometrical structure of such a bed. Bars with diameters were of 10, 20, 30, and 40 mm made of S 235JRH steel were used in the tests, while the gas phase of the bed was air.

The measurement principle involves forcing a one-dimensional steady heat flux *q* between the bottom and top surfaces of the sample. After reaching a steady state, the temperatures on the bottom surface *t_bo_* and the top surface *t_to_* are measured. Similar to the definition of solid material thermal conductivity *k_s_* the effective thermal conductivity is defined as in (Equation (5)) [18,19]:(5)$$k_{ef}=\frac{q\cdot l_{sp}}{t_{bo}-t_{to}}=\frac{q\cdot l_{sp}}{\Delta t},$$
where *l_sp_* is the dimension of the sample in the direction of the heat flow. This parameter depends on the number of bar layers in the sample, its arrangement, and the bar diameter.

A custom experimental stand was used for the measurements. A general view of this stand is shown in Figure 3, which consisted of a heating chamber, a sample temperature measurement system, a control system for the main heater power supply, a control system for the guarded heaters, and a cooling system. The main component of this stand is the heating chamber, which is schematically shown in Figure 4.

The tested samples are placed on the bottom of a rectangular retort, with internal dimensions for the base of 400 × 400 mm and a height of 200 mm, made of a 4 mm boilerplate. From the top, the retort end has a flange with a cover fixed with bolts. There is a main heater under the retort bottom, with the same transverse dimensions of 400 × 400 mm. The heat generated in the main heater is entirely transferred towards the tested sample. Fulfilling this condition is ensured by the two guarded heaters (the side one and the bottom one). The power of the main heater is adjusted manually using an autotransformer. This allows control of the mean measured temperature. Furthermore, the power of the guarded heaters is adjusted automatically using a special control system. To limit the undesired heat loss from lateral surfaces of the heaters and the retort, the heating chamber was isolated using the ceramic fabric.

During the tests in the guarded hot plate apparatuses, the bottom and top surfaces of the sample were usually in contact with the main heater and the cooler, respectively. This solution was not used for the discussed test stand. If the cooler was in direct contact with the top surface of the samples, its effect would limit the temperature range of the measurement. The examinations aimed to determine the thermal conductivity for the greatest possible temperature range. The water cooler was used but it was installed in the chamber covers, and consequently it was located away from the surface of the sample. Thanks to this solution the cooler did not lower the temperature significantly; however, at the same time it forced a one-dimensional heat flow.

The temperature measurements on the sample surfaces were carried out using TP-201 0.5 mm K-type sheathed thermocouples [20] connected to a multichannel data logger equipped with an EMT 200 temperature meter [21]. Temperatures on the bottom and top surfaces were measured at five opposite points. One point was located in the geometrical center of the surface, whereas the four other points were located in the corners of the square, with sides measuring 260 mm and the center being coincident with the sample center. The thermocouples used for temperature measurement at the bottom surface of the sample were fixed to the retort bottom that served as the hot plate. Furthermore, the thermocouples used for temperature measurement on the top surface of the sample were fixed to the steel plate that covered the samples. Due to the cooler shift, this plate performed the role of the cold plate. Its thickness was 15 mm, while its transverse dimensions measured 390 × 390 mm.

Heat flux *q* was evaluated as a quotient of heat flux rate *Q* generated in the main heater and its surface area *A*. The value of *Q* was assumed to be equal to the current power *P* supplied to this heater. This approach is possible because electric resistance heaters are typically 100% efficient, which means that all of the electrical energy used is converted into heat [22]. The current power *P* was measured using an N14 3-phase power network meter [23]. The measurements for each sample were made for different adjustments of the current power *P*, which changed from 200 to 3200 W. This procedure allowed for determination of the temperature changes in the value of coefficient *k_ef_*.

The key element of the experimental investigations is the analysis of the uncertainty of measurement. For the investigation of the effective thermal conductivity with the use of a guarded hot plate apparatus, the relative uncertainty of measurement is estimated from an error propagation rule [24]. The equation used to calculate the relative uncertainty of measurement, in this case, takes the following form (Equation (6)):(6)$$\frac{\delta k_{ef}}{k_{ef}}=\sqrt{{\left(\frac{\delta P}{P}\right)}^{2}+{\left(\frac{\delta A}{A}\right)}^{2}+{\left(\frac{\delta l_{sp}}{l_{sp}}\right)}^{2}+{\left(\frac{\delta \Delta t}{\Delta t}\right)}^{2}},$$

Individual terms from the left-hand-side of Equation (6) denote the relative measurement uncertainties for the current power of the main heater *P*, the surface area of the main heater *A*, the dimension of the sample *l_sp_*, and the temperature difference Δ*t* between the lower and upper surface of the sample. This means that the relative measurement uncertainty of the *k_ef_* coefficient depends on the relative uncertainty of measurement of four values: *P*, *A*, *l_sp_*, and Δ*t.* According to the information contained in the technical specification for the wattmeter, the uncertainty of the measurement of the main heater power *P* is 2% [23]. The surface area of the main heater *A* was set based on the measurement of the length of its sides. This measurement was made with an accuracy of 1 mm, which for the surface *A*, yielded uncertainty of 0.4%. The measurement of the sample dimension *l_sp_* was performed using the slide calliper, with an accuracy of 0.5 mm. With consideration of the sample dimension, the maximal uncertainty of this measurement was 0.8%. According to the manufacturer’s data, the measurement of the temperature was characterized by an uncertainty level of 0.5 °C [21]. Consequently, the maximal uncertainty of the measurement of the temperature difference Δ*t* was 4%. Finally, taking into account the enumerated constituent uncertainties, the maximal measurement uncertainty of the effective thermal conductivity for the discussed stand was 4.6%.

The effective thermal conductivity measurement results obtained for the described samples are presented in Figure 5. Individual points obtained for a given sample refer to the subsequent settings of the current power of the main heater. As can be seen, the coefficient *k_ef_* depends on the diameter of the bars and increases linearly with temperature, making it possible to approximate the results for each sample with the following equation (Equation (7)):(7)$$k_{ef}=B_{1}+B_{2}t,$$

Table 1 presents parameters *B*_1_ and *B*_2_, as well as the minimum and maximum of the *k_ef_* coefficient in the temperature range of 25–700 °C for specific diameters of the bars. The goodness of fit of Equation (7) to the measurement results is expressed with the coefficient of determination *R*^2^, the values of which are presented in Table 1. The values of *R*^2^ close to 1 indicate a good match.

Other values necessary for calculating the *f* coefficient based on Equation (4) are the heat conductivities of the solid and gas phase of the analyzed medium. When performing calculations, it was assumed that the material of the solid phase of the analyzed medium was S 235JRH low-carbon steel. Bars made of steel of this kind were used in the experimental investigation of the effective thermal conductivity. Changes of thermal conductivity for this kind of steel regarding the temperature function are described by Equation (8) [25]:(8)$$k_{s}=1.2\cdot 10^{-8}t^{3}-3.2\cdot 10^{-5}t^{2}-1.2\cdot 10^{-2}\;t+51.3,$$

The gas phase of a bar bed is air, the thermal conductivity *k_g_* of which depending on temperature can be described by Equation (9) [25]:(9)$$k_{g}=-2.88\cdot 10^{-8}t^{2}+8.05\cdot 10^{-5}\;t+0.024,$$

When analyzing heat transfer in a bar bed, one must be aware that in the voids, along with conduction in the air, the heat is also transferred as a result of radiation between the surfaces of adjacent bars. It has been shown that in the temperature range of 25–800 °C, approximately 30% of heat within the bar bundle is transferred through radiation [26]; therefore, we decided to take this kind of heat transfer into account in the analyses. The influence of thermal radiation was taken into account by substituting the thermal conductivity of gas *k_g_* in Equation (4) with a replacement void thermal conductivity *k**_ϕ_*, which was defined in the following way (Equation (10)) [27,28]:(10)$$k_{}=k_{g}+k_{rd},$$
where the *k_rd_* coefficient indicates radiation thermal conductivity, which quantitatively expresses the heat transferred in the porous medium as a result of radiation. The value of the *k_rd_* coefficient for a bar bundle can be calculated using the radiosity balance method [29]. The starting point for this analysis is the geometric model of the considered medium. This is a repeatable fragment of a bed of bars in a staggered arrangement (Figure 6a). The basis for deriving all mathematical relations was extracted from this fragment of the elementary cell (Figure 6b), in which vertical heat flow takes place, described quantitatively by the heat flux *q*. For the performed calculations, the cell is divided into three vertical sections, a central section (*II*) and two identical side sections (*I*, *III*). To define the cell geometry, the bar diameter *d_b_* and the width of the gap between the bars from a single layer *l_gp_* are used. Based on these two parameters, the cell height *δ_K_*, width of the sections, and bundle porosity *ϕ* are determined.

In the cell area there are four surfaces (*A*_1_, *A_I_*, *A_III_*, and *A_gp_*), between which radiation heat transfer occurs. The thermal resistance for radiation *R_rd_* in this system is described by Equation (11) [30]:(11)$$R_{rd}=\frac{X_{rd}}{4\sigma T^{3}},$$
where *X_rd_* is a dimensionless coefficient, the value of which depends on the bar emissivity *ε* and its shape, as well as the relative position of the surfaces that close the space of radiation exchange; *σ* is the Stefan–Boltzman constant; *T* is the absolute mean temperature of all surfaces. For the arrangement of the surfaces in the considered cell coefficient, *X_rd_* is described by Equation (12) [26]:(12)$$X_{rd}=\frac{2l_{p}+l_{ga}}{\left(1.59d_{b}-2A_{I}\right){\left(\frac{1}{\varepsilon }+\frac{A_{I}}{2A_{I}+A_{gp}}\left(\frac{1}{\varepsilon }-1\right)\right)}^{-1}},$$

As a result of rearranging Equations (11) and (12), which are described in detail in [31], it is possible to obtain a dependence that describes the value of the coefficient *k_rd_* for the analyzed medium (Equation (13)):(13)$$k_{rd}=4\left[\left(-2.58\varphi +1.14\right)\varepsilon +0.96\;\varphi -0.29\right]\sigma d_{b}T^{3},$$
where *T* is the absolute mean temperature of the medium. The form of Equation (13) results from a commonly applied relationship that is used to determine the radiation thermal conductivity in porous media built from spherical or cylindrical particles (Equation (14)) [3,32]:(14)$$k_{rd}=4F_{R}\sigma d_{b}T^{3},$$
where *F_R_* is a radiation exchange factor.

When calculating the value of the *k_rd_* coefficient according to Equation (13), it was assumed that the bar emissivity grows in the temperature function. The changes of emissivity in the temperature function were described by the experimentally established relationship (Equation (15)) [33]:(15)ε=0.64+0.0002 t,

Table 2 presents the values of the *k_s_*, *k_g_*, and *k_rd_* coefficients (depending on the bar diameter) for the chosen temperatures.

When making the calculation, it was also assumed that the porosity of the analyzed charge was constant and equaled 0.1. This value corresponds to the staggered arrangement of the bars with maximum packing.

## 3. Results and Discussion

The calculation results for the *f* coefficient from the Krischer model for the considered bar bundles are presented in Figure 7. The value of this parameter depending on the analyzed case is varied and generally ranges from 0.0845 to 0.3201. Table 3 presents the minimum, average, and maximum values of the *f* parameter obtained for individual bar diameters. This value decreases with increases in the bar diameter, whereas changes in the temperature can be approximated with the third-degree polynomial (Equation (16)):(16)$$f=C_{1}t^{3}+C_{2}t^{2}+C_{3}t+C_{4},$$

Table 4 presents the *C*_1_*-C*_4_ coefficients from Equation (16) determined for individual bars and the *R*^2^ coefficient. The values of *R*^2^ close to 1 are a sign of a good match of the results obtained with the use of Equation (16) and the results of model calculations (Equation (4)).

The results of the *f* parameter calculations indicate that the Krischer model can be used to determine the effective thermal conductivity of the bundles of steel bars; however, obtaining correct results requires precise matching of changes of the *f* parameter, both in the function of the temperature and the bar diameter. This solution, called the full solution, enables the use of Equation (16) and the coefficient values, which are collated in Table 4; this approach is not very practical though. For this reason, we additionally checked which results would be obtained with the use of the Krischer model for constant values of the *f* parameter when averaged for temperature (these values are collated in Table 3). This approach is called a simplified solution. The effective thermal conductivity obtained with the use of the simplified solution of the Krischer model (Figure 8) differed from the values described by Equation (7). Above all, *k_ef_* changes in the temperature function were not linear in this case. The intensity of growth decreased with the temperature increase. For diameters equal to 30 and 40 mm at temperatures above 400 °C, there was even a slight decrease of *k_ef_*.

We will now discuss the obtained results by explaining the influence of the bar diameter on the value of the *k_ef_* parameter. The experimental measurements in Figure 5 show that the effective thermal conductivities depend on the bar diameter. The same behaviour can be observed for the results of the Krischer model in Figure 8. Similar trends were reported in the literature, for example for thermal conductivity of polyamide–carbon nanotube composites [34]. These observations can be explained as follows. The *k_ef_* coefficient refers to the distance in which heat transfer occurs, which is shown in meters (a unit of length). Heat transfer in the bar bed can be analyzed with the use of the notion of thermal resistance. In simple terms, the total thermal resistance of the bar bed *R_to_* should be represented as the series combination of two resistances: the conduction resistance of the bars *R_b_* and the thermal resistance of the gap *R_gp_* [30]. The *R_gp_* resistance is much greater than the *R_b_* resistance. When the bar diameter is smaller, there are more layers on the unitary length of the bed, meaning more *R_gp_* resistance. This explains the influence of the bar diameter on the *k_ef_* value—*k_ef_* grows with the increase of the diameter because there is a smaller number of *R_gp_* resistances per unit of length. Additionally, the contribution of radiation to heat transfer increases with the bar diameter, which manifests itself with the increased value of the *k_rd_* coefficient (as shown in Table 2). This effect results directly from Equation (14), according to which the value of *k_rd_* is in direct proportion to *d*_p_. This is because the increase of the bar diameter increases the distance between the surfaces of the radiation heat transfer (surfaces *A*_1_, *A_I_*, *A_III_*, and *A_gp_* in Figure 6b), with the result being that in the constant temperature field, the heat flux of the same value is transferred further away. Because the *k_rd_* coefficient (according to the definition of its unit) refers to a distance of heat transfer, the growth of *d*_p_ increases the value of *k_rd_*. It is necessary to be aware that the presented interpretation is considerably simplified; however, we consider it is sufficient to explain the observed relationship. A more detailed analysis of this problem will be the subject of future publications by our group. 

To quantitatively determine the divergence between the results of the full solution *k_ef-f_* and the simplified solution *k_ef-s_*, the relative percentage difference of the effective thermal conductivity *dk_ef_* was used, which was defined in the following way (Equation (17)):(17)$$dk_{ef}=\frac{k_{ef-f}-k_{ef-s}}{k_{ef-f}}100\%.$$

Figure 9 presents the calculation results for the difference *dk_ef_*. The values of this parameter range from approximately −21% to 32%. A particularly high proportion occurs within the temperature range of 25–200 °C and 600–800 °C. Keeping in mind the correct method for heating a bar bundle, the obtained results lead to the conclusion that the simplified Krischer model cannot be applied in industrial practice. The feasible solution is the full solution, in which the value of the *f* coefficient depends both on the bar diameter and temperature; therefore, despite the simple formula of the model itself, as described with Equations (1)–(3), its practical application in the discussed situation is not easy.

## 4. Conclusions

The correct control of the heat treatment process of bundles of steel bars requires precise knowledge of the thermal properties. As a bundle is a porous medium, its basic thermal property is the effective thermal conductivity, which differs significantly from the thermal conductivity of steel. The possibility of relatively simple prediction of the *k_ef_* coefficient changes of the heated bundle, both in the temperature function and bar diameter, is of key importance in industrial practice. It has been demonstrated that using the Krischer model for this aim requires careful estimation of the *f* coefficient. In cases of missing information for the *k_ef_* value, the *f* estimation is practically impossible. As a result, despite the simple mathematical notation, the analyzed model could not be successfully implemented in the case discussed in this paper.

The main advances of this paper are presenting the results of measurements of the effective thermal conductivity of bundles of steel bars, presenting the formula that enables radiation to be considered in the Krischer model, determining the value of the weighting parameter *f* for which the Krischer model enables one to determine the thermal properties if a bar bundle, and determining the uncertainties that are characteristic of the calculation results of the Krischer model for a simplified solution.

## Figures and Tables

**Figure 1 materials-14-04378-f001:**
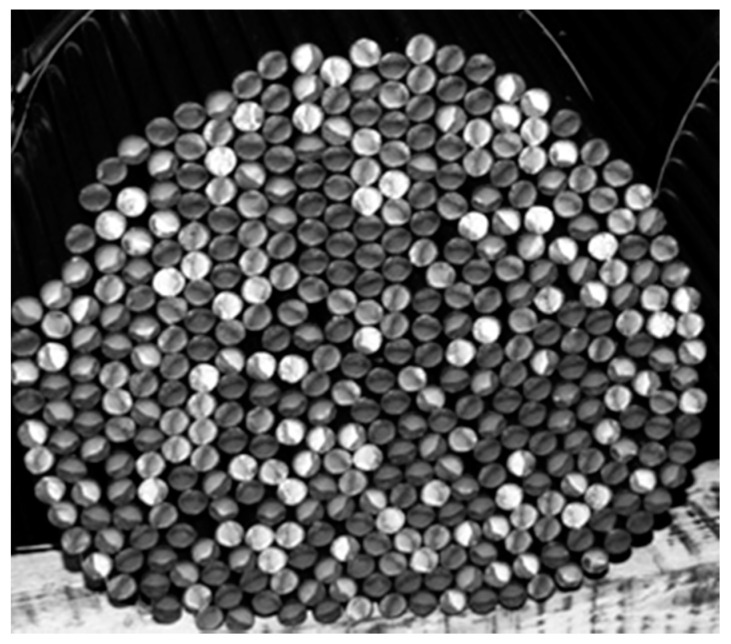
A view of a bar bundle prepared for heat treatment, which shows the cellular structure of the charge.

**Figure 2 materials-14-04378-f002:**
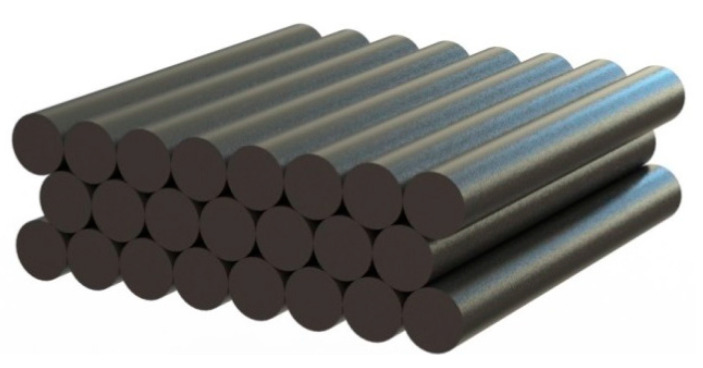
Scheme of the geometrical structure of a flatbed of round bars with a staggered arrangement.

**Figure 3 materials-14-04378-f003:**
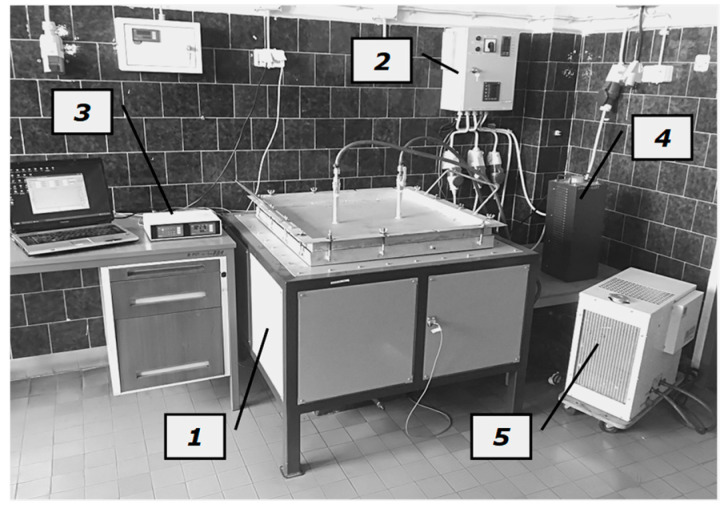
A general view of the testing stand: 1—heating chamber; 2—control unit for the main and guarded heaters; 3—data logger with temperature meter; 4—autotransformer; 5—cooling system unit.

**Figure 4 materials-14-04378-f004:**
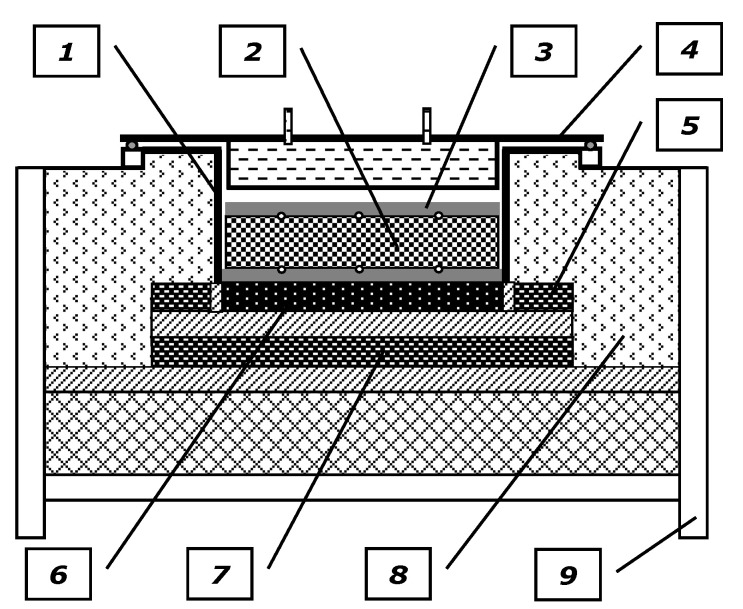
Scheme of the heating chamber: 1—retort with a hot plate; 2—investigated sample; 3—cold plate; 4—heating chamber cover with a cooler; 5—side-guarded heater; 6—main heater; 7—bottom-guarded heater; 8—thermal insulation; 9—support structure.

**Figure 5 materials-14-04378-f005:**
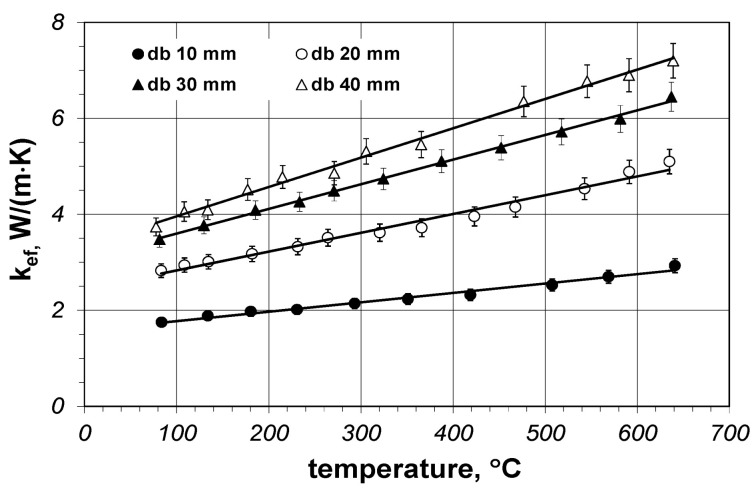
Measured effective thermal conductivity vs. temperature for flat packed beds of steel bars.

**Figure 6 materials-14-04378-f006:**
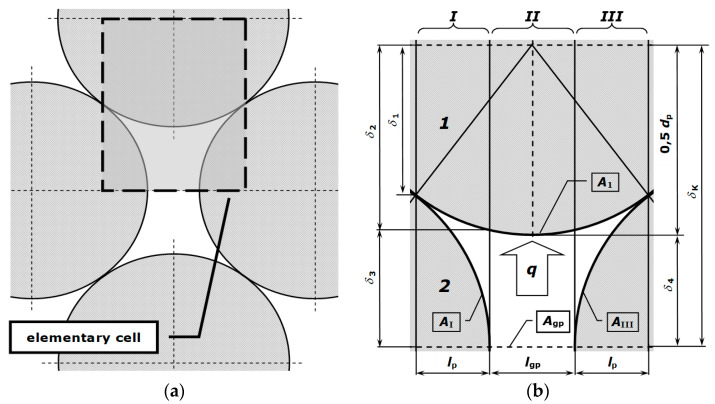
The elementary cell: (**a**) part of the considered charge; (**b**) elementary cell used for thermal resistance analysis.

**Figure 7 materials-14-04378-f007:**
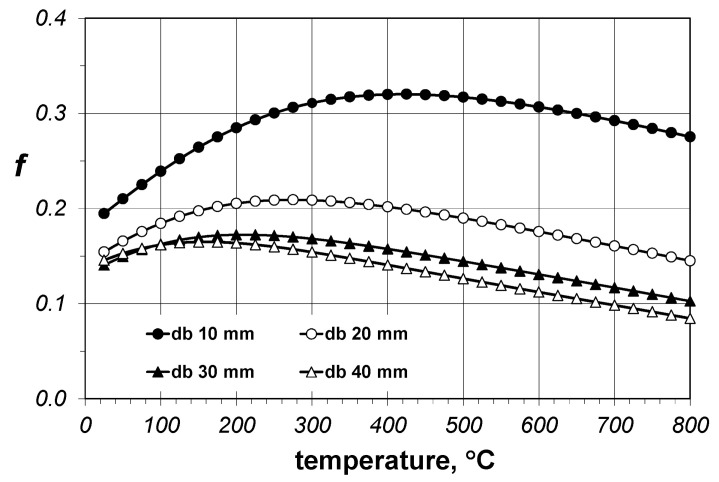
The values of the *f* coefficient depending on the bar diameter and temperature.

**Figure 8 materials-14-04378-f008:**
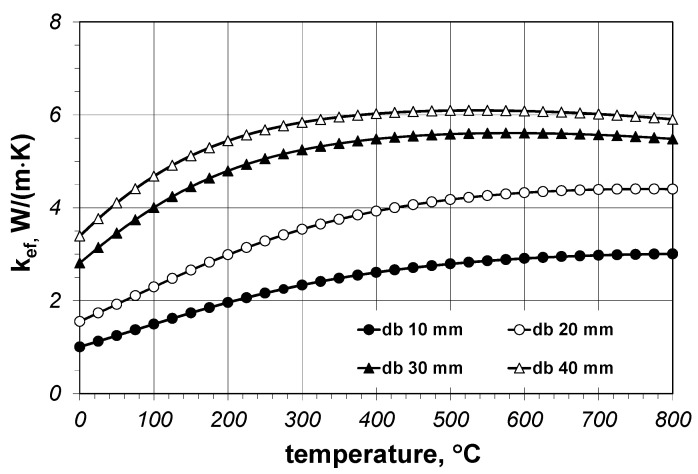
The results of calculations of the effective thermal conductivity *k_ef_* obtained with the use of the Krischer model for a simplified solution.

**Figure 9 materials-14-04378-f009:**
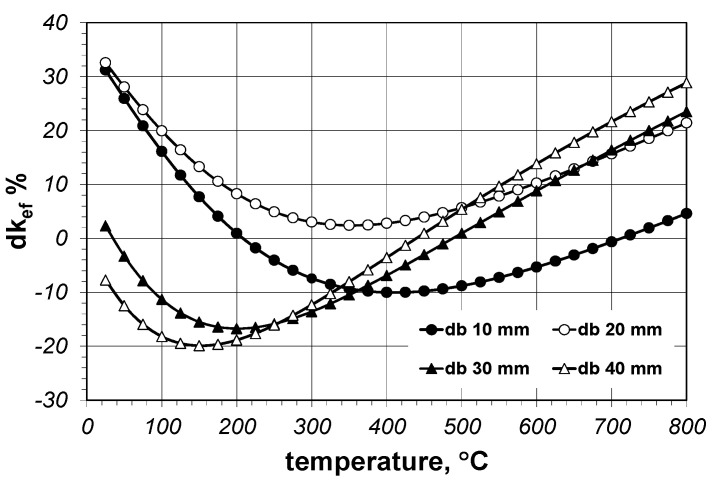
The results of calculations of the relative percentage difference *dk_ef_* between the full solution and the simplified solution.

**Table 1 materials-14-04378-t001:** The values for *B*_1_, *B*_2_, *R*^2^, *k_ef-min_*, and *k_ef-max_* coefficients obtained for the individual diameters of the bars.

*d_b_*, mm	*B* _1_	*B* _2_	*R* ^2^	*k_ef-min_*, W/(m K)	*k_ef-max_*, W/(m K)
10	1.59	0.0019	0.981	1.63	2.96
20	2.48	0.0039	0.974	2.58	5.21
30	3.09	0.0051	0.998	3.22	6.65
40	3.34	0.0062	0.987	3.49	7.68

**Table 2 materials-14-04378-t002:** The values of *k_s_*, *k_g_*, and *k_rd_* coefficients for the chosen temperatures.

*t*, °C	*k_s_*, W/(m K)	*k_g_*, W/(m K)	*k_rd_*, W/(m K)			
			*d_b_* 10 mm	*d_b_* 20 mm	*d_b_* 30 mm	*d_b_* 40 mm
25	50.3	0.024	0.017	0.035	0.052	0.069
200	47.7	0.039	0.098	0.197	0.295	0.394
400	42.1	0.051	0.309	0.617	0.926	1.235
600	35.1	0.062	0.729	1.457	2.186	2.914
800	27.3	0.069	1.454	2.908	4.362	5.817

**Table 3 materials-14-04378-t003:** Minimum, average, and maximum values of the *f* parameters depending on the bar diameter.

*d_b_*, mm	*f_min_*	*f_av_*	*f_max_*
10	0.1789	0.2862	0.3201
20	0.1421	0.1832	0.2091
30	0.1026	0.1458	0.1724
40	0.0845	0.1333	0.1649

**Table 4 materials-14-04378-t004:** The *C*_1_–*C*_4_ coefficients from Equation (16), depending on the bar diameter.

*d_b_*, mm	*C* _1_	*C* _2_	*C* _3_	*C* _4_	*R* ^2^
10	5.71 × 10^−10^	−1.271 × 10^−6^	7.78 × 10^−4^	0.176	0.999
20	6.55 × 10^−10^	−1.151 × 10^−6^	5.03 × 10^−4^	0.144	0.996
30	5.77 × 10^−10^	−0.938 × 10^−6^	3.46 × 10^−4^	0.134	0.992
40	5.21 × 10^−10^	−0.797 × 10^−6^	2.38 × 10^−4^	0.142	0.992

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
