# Peer review of "On Determination of the Effective Thermal Conductivity of a Bundle of Steel Bars Using the Krischer Model and Considering Thermal Radiation"

_materials, 2021, doi:10.3390/ma14164378_

Round 1

Reviewer 1 Report

Cellular solid materials are kind of porous structure materials which were applied in many engineering fields and it is important to study its thermal properties. The current work was to investigates the effective thermal conductivity of heat treated bundles of steel bars based on the results of experimental tests and performed calculations, using the Krischer model and taking into account thermal radiation. Generally, the topic seems interesting and some problems should be addressed.

1. The organization of the paper could be improved, e.g., the part including Eq.(1) to Eq.(3) is suggested to be adjusted to the Part 2.

2. Do Fig.2 just present the structure?

3. The conclusion part is suggested to be improved to highlight the main advantages.

Reviewer 2 Report

The authors have revised the paper according to reviewers' suggestions. The results are now clearly present. 

I recommend the authors to revise the language carefully and make sure the variables are consistent through the paper. 

Reviewer 3 Report

In this manuscript WyczóÅ‚kowski et al. have done experiments on the thermal conductivity of bundles of metal bars and provided an analysis based on the Krischer model to examine the temperature- and tube diameter-dependence of the thermal conductivities. They have further developed the Krischer model to include the radiative heat transfer (through voids between metal bars). The paper is well written and understandable and the conclusion are supported by the results. However, I have a technical comment, to be considered before publication:

The experimental measurements (Fig. 5) show that the thermal conductivities depend on the diameter of the metal bar. The same behaviour is observed in the modeling based on the Krischer model. This is reasonable, increasing the diameter of the metal bar provides a larger surface area of contact between them, and hence, increases the rate of heat transfer; two infinite-diameter tube (two flat surfaces) are in complete contact. The same trend has been reported in the literature for heat conduction across nanotubes and flat graphene surfaces (see for example Polymers 2019, 11, 1465 & J. Chem. Phys. 2011, 135, 064703). However, on the other hand, the relative surface area which is not in contact (Figure 6), and therefore contributes to heat flow via radiation decreases with increasing the tube diameter. But the results in Table 2 show that the contribution of radiation to heat transfer also increases with tube diameter. In my opinion the modifications (improvements) for inclusion of the role of radiation in heat transfer do not reflect the correct diameter dependency. Please comment on this point in the manuscript.

Round 2

Reviewer 1 Report

The problems I had proposed in the last version was carefully checked and revised. The paper seems fine and I recommend its publication. 

Reviewer 2 Report

The paper has been improved. I recommend it to be accepted. 

Reviewer 3 Report

All my comments are taken into account. I recommend publication.

This manuscript is a resubmission of an earlier submission. The following is a list of the peer review reports and author responses from that submission.

Round 1

Reviewer 1 Report

Bundles of bars are cellular structures which is applied in many industrial applications, and it is a kind of granular two-phase medium. It is important to investigate its thermal property. Up to now, several analytical models were developed for determining the effective thermal conductivity of the two-phase media. The current paper is to determine of the effective thermal conductivity of a bundle of steel bars using the Krischer model taking into account thermal radiation. Generally, the topic seems interesting, and some problems should be addreed. 1. It is suggested the author could highlight the main advances of the current paper. 2. The organization of the paper could be modified, e.g., the introduction part. 3. Generally, the results are to obtained through experiment , which should be carefully could be checked

Reviewer 2 Report

In this paper, the authors discussed the application of the Krischer model to describe the thermal conductivity of a bundle of steel bars. By using a constant f value, the relative differences of the thermal conductivity calculated using the Krischer model and measured in the experiments range from -21% to 32%. Therefore, the authors concluded that the Krischer model is not valid to determine the thermal conductivity of a bundle of steel bars. However, the major limitation of this paper is that the discussions and validations in the paper are based on a set of experimental measurements published by the authors in 2017, but not available to general audience. There are no details for the reviewers and readers to judge the measurements. The authors should validate their results with other measurements that are available in the literature and conducted from other groups.

Reviewer 3 Report

Thank your for your review invitation of your manuscript. This is my recommendations.

  1. In your caculation, thermal resistance will be very important, because basically your model is induced from resistance network model between bars. But I could not catch what your modl is.
  2. You should present your calculation model with your approach by figure or sketch etc. to explain your physical approach.  Figure 1 is just photo, it is not explain your approach.
  3. Physical geomentry or heat flow path will be improtant to explain your model.
  4. You mentioned and present value of R2, but it is not explained how that varable is used in equation or others.
  5. Equation 9-11 is not correct equations for circular bar. Reference 18 shows those equations with rectangular hollow bar. That means that you have to present your verification of these equatipons.
  6. Please define your variables in nomenclature, few symbols of your equatipons are not defined.
  7. You should explain why the model of Eq. (1) is adopdted in your research. It is not clear.
  8. In your manuscript, it is not clear what you want to explain or claim from your research.